# Contrasting Physical and Virtual Museum Experiences: A Study of Audience Behavior in Replica-Based Environments

**DOI:** 10.3390/s25134046

**Published:** 2025-06-29

**Authors:** Haojun Xu, Yuzhi Li, Feng Tian

**Affiliations:** Shanghai Film Academy, Shanghai University, Shanghai 200072, China; sacross_k@shu.edu.cn (H.X.); shadowmcv@shu.edu.cn (Y.L.)

**Keywords:** virtual museum, multi-user mixed-reality, audience behavior, path analysis

## Abstract

This study explores the differences in audience behavior between virtual museums and physical museums. The replica-based virtual museum (RVM) was developed to replicate the exhibit layout of physical museums and support multi-user online visits. The study introduces the RVM-Interaction (RVM-I), which incorporates interactive features to enhance user engagement. In the experiment, 24 participants experienced a physical museum (PM), RVM, RVM-I, and a traditional PC-based virtual museum, with their impressions and behavioral patterns recorded. The results indicate no significant differences between RVM and PM in terms of satisfaction, immersion, aesthetic experience, and social interaction. RVM-I significantly enhanced the participants’ experience through its interactive capabilities. Path analysis shows that both RVM and RVM-I improved audience efficiency, with RVM-I transforming the circumferential, space-based art appreciation found in PM and RVM into a stationary, space-based form, making RVM-I more engaging than RVM. These findings offer valuable insights for the design and development of virtual museum experiences that maintain spatial fidelity to physical exhibitions while enhancing user engagement through interactivity.

## 1. Introduction

With people’s increasing interest in museum exhibitions [1], physical museums (PM) are seeing a surge in visitor numbers. However, high visitor traffic often hinders individuals from closely observing exhibits, negatively impacting the overall experience (Figure 1, left). In response, virtual museums have emerged as a significant medium for cultural dissemination, providing a new way for people to engage with historical artifacts [2,3]. Compared to traditional museums, virtual museums offer advantages such as digitization, interactivity, and the elimination of geographic barriers, allowing audiences to access exhibits at any time and from any location [4].

The recent research on virtual museums has primarily focused on exhibition formats [5,6,7], tour guides [8,9,10], and enhancing visitor experiences [11,12], with many studies emphasizing the use of handheld display (HHD) devices. However, there has been limited analysis of behavioral differences between audiences in fully recreated virtual museums and those in physical museums, making it difficult to fully understand variations in how audiences view artifacts and navigate exhibition spaces.

To address this issue, this paper introduces the replica-based virtual museum (RVM), a highly realistic virtual museum system that replicates the physical museum environment and supports multi-user visits (Figure 1, right). RVM is designed to offer a museum experience comparable to that of a physical visit. Furthermore, we developed an extended system, replica-based virtual museum-interactable (RVM-I), which incorporates interactive features, allowing for the analysis of how interactivity in virtual museums [13] influences visitor experiences.

This study is driven by practical challenges in physical museums—including overcrowding, limited exhibit visibility, and restricted interaction [14,15,16]—which hinder meaningful cultural engagement. To explore how virtual museum replicas might address these issues, we developed RVM and RVM-I, aiming to preserve the authenticity of physical spaces while removing crowd-related constraints. We investigate two key questions:How does the experience in RVM compare to that of a physical museum?Can interactivity in RVM-I provide a more engaging alternative when physical visits are constrained or impractical?

We selected the bronze exhibition area and the Sanxingdui exhibition area from a renowned museum as our experimental environment, faithfully replicating the exhibits and exhibition configuration in the RVM system. To overcome challenges such as low lighting, the inability to touch artifacts, and limitations on photography in physical museums, we developed a virtual reality (VR) stereoscopic photography guidance system. By synthesizing multiple images, we generated high dynamic range (HDR) binocular visuals [17,18] and employed multi-view stereo reconstruction technology [19,20] to create accurate 3D models of the artifacts, aiming to closely replicate the physical exhibition space in the virtual museum.

We recruited 24 participants to experience the physical museum, RVM, RVM-I, and a traditional PC-based virtual museum (as a control group) and recorded their impressions and behavioral patterns. The experimental results showed that in terms of satisfaction, immersion, aesthetic experience, and social interaction, RVM closely mimics the physical museum, with no significant differences. This suggests that virtual museums can, to some extent, replicate the experience of visiting physical museums.

RVM-I, through its interactive features, significantly enhanced participants’ experiences, highlighting the advantages of virtual museums in delivering interactive experiences. Although the overall time spent in RVM was shorter than in the physical museum, the time spent closely observing artifacts was comparable. In RVM-I, the time spent viewing artifacts increased significantly compared to the other two conditions.

This work not only advances the technical capabilities of virtual museums but also broadens the discourse on how mixed reality can bridge the gap between physical and digital heritage experiences.

The main contributions of this paper are as follows:We present the replica-based virtual museum (RVM), a novel multi-user mixed reality (MU-MR) system that replicates physical museums at a 1:1 scale. It enables rapid setup in open spaces without dedicated infrastructure, offering a flexible solution for virtual cultural experiences.We extend RVM with RVM-I, incorporating gesture-based interaction to enhance immersion and user engagement, demonstrating the value of interactivity in MU-MR museum environments.A user study comparing physical museums, PC-based virtual museums, and our systems shows that RVM offers comparable satisfaction, immersion, aesthetics, and social interaction, validating its effectiveness.Behavioral analysis reveals that RVM improves tour efficiency, while RVM-I encourages focused observation over exploration, highlighting how interactivity reshapes engagement and learning in virtual museums.

## 2. Related Work

### 2.1. Virtual Museum

As a product of the digital age, virtual museums have become an important method for preserving and disseminating cultural heritage. As early as the late 20th century, Tsichritzis et al. introduced the concept of the virtual museum, aiming to overcome the limitations of traditional physical museums and provide vivid experiences for remote audiences [21]. Increasingly, museums are investing in the development of virtual museums with the goal of broadening cultural dissemination and educational outreach. The primary form of virtual museums today includes text and image displays of exhibits and 360-degree panoramic views, enabling global audiences to access collections remotely [22,23]. Google’s Art Project, currently the largest virtual museum initiative, collaborates with museums worldwide, using Google Street View technology and ultra-high-resolution photography to display a vast range of artworks [24]. Similarly, the British Museum has launched its own virtual museum project, allowing global audiences to explore its collections online [25].

In applied research, there has been substantial focus on the practical uses of virtual museums. Marcello et al. found that using VR for museum displays can increase viewer engagement [26]. Sylaiou et al. suggested that virtual museums offer new educational pathways, promoting global cultural exchange and integration through online exhibitions and remote teaching [27]. Additionally, VR-based reconstructions for museum tours are emerging, creating more immersive and realistic experiences [10,26,28,29].

Augmented reality (AR) technology is mainly applied to museum guidance systems, with applications typically relying on HHD devices to assist audiences during tours [8,9]. AR is also used to enhance real-world information, such as the restoration of historical buildings and landscapes [30,31].

Most studies on virtual museums rely on surveys to gather audiences feedback, with fewer studies focusing on visitor traffic. Marín-Morales et al. [32] analyzed differences in visitor paths between a virtual museum and a physical museum. However, because the virtual museum space exceeded the typical size for HTC VIVE, audiences in the virtual environment could only move using the teleportation method, limiting their ability to fully replicate a physical museum experience. The ViewR system enabled large-scale MU-MR virtual museums, but its path statistics focused primarily on system stability without comparison to physical environments [33].

In summary, virtual museums remain a prominent area of academic research. With advancements in head-mounted displays (HMD), the focus of virtual museum research has gradually shifted from early image displays to using MR for exhibitions. However, current MR applications predominantly use HMD devices for virtual museum tours, with limited comparative studies examining the exhibition and movement patterns between MU-MR virtual museums and physical museums.

### 2.2. MU-MR Application

Over the past two decades, MR technology has garnered significant attention from both academia and industry. With the introduction of video see-through capabilities in consumer-grade HMDs, research and applications have increasingly focused on MU-MR systems. Compared to MU-VR systems, MU-MR offers audiences a more intuitive interaction with real-world environments, allowing seamless integration of virtual objects with physical spaces, thus providing substantial advantages. Local collaboration in MU-MR systems fosters more natural communication, offering benefits like instant feedback and easier interaction compared to remote collaboration [34,35].

Ryan Anthony J. de Belen et al. reviewed 259 papers on collaborative MR published between 2013 and 2018, categorizing them by research area, devices used, and types of collaboration [34]. The study highlighted the advantages of seamless interaction between virtual and real environments enabled by MR, particularly in industrial collaboration. Of these papers, 65 involved HMDs such as Oculus Rift, HoloLens, and HTC VIVE. Additionally, 103 papers focused on local collaboration, while 27 supported both local and remote collaboration. Notable examples include the multi-user painting interaction system, CollaboVR [36], and the multi-user character animation creation system, PoseMMR [37]. While systems like HTC VIVE handle virtual space synchronization through spatial positioning, they are still limited by the range of the positioning system.

The development of visual SLAM technology [38] has enabled “inside-out” positioning systems, allowing newer commercial-grade HMDs to perform markerless spatial positioning with greater accuracy than “outside-in” systems [39,40]. This advancement overcomes previous spatial limitations and improves user experiences. However, such systems also present challenges in synchronizing the virtual spaces of multiple users within the same physical environment. The cross-platform xR system proposed by [41] addresses this issue, allowing multiple devices to collaborate locally. For multi-device calibration, HoloLens 2 is used to scan objects from multiple angles, aligning the point cloud data with the virtual space in the HoloLens. Florian Schier et al. developed the ViewR system, a large-scale multi-user MR spatial exhibition system, which uses gestures or controllers to calibrate spatial coordinates. This system demonstrated the feasibility of large-scale spatial MR using commercial-grade HMDs [33].

In summary, MU-MR systems significantly enhance user interaction and provide more intuitive experiences. However, most systems still rely on computers for rendering, limiting user mobility due to hardware constraints. Although the latest generation of commercial-grade HMDs has addressed some hardware limitations and increased mobility, the calibration process remains complex, often requiring pre-arranged environments, which limits the system’s general applicability.

## 3. Replica-Based Virtual Museum

### 3.1. Material

Bronze, one of the earliest alloys invented by humans, is regarded as a symbol of civilization and held a sacred status in China for many centuries [42]. In this study, we selected the bronze exhibition hall as our research environment, employing various techniques to construct a virtual museum that closely replicates the real exhibit displays.

During the digitization process, we applied multi-view stereo (MVS) reconstruction techniques based on close-range photogrammetry principles to generate 3D models of the artifacts with high visual fidelity [43,44,45]. Stereoscopic images were captured from four cardinal viewpoints around each object, and depth maps were computed using stereo matching algorithms [28]. These were merged into a unified point cloud dataset, which served as the basis for surface reconstruction. A triangular mesh was generated and further refined through post-processing steps including mesh simplification, hole filling, and texture mapping to ensure rendering performance and visual realism. To enhance spatial consistency without relying on precise camera parameters (typically obtained via drones or specialized equipment), our VR-based guidance system was used to manually align viewpoints to predefined positions, effectively replacing the initial camera pose estimation typically performed by structure-from-motion (SfM) algorithms [46,47]. This approach circumvents the need for high-accuracy camera calibration, instead prioritizing visual alignment and geometric plausibility through manual intervention.

In practice, the museum’s low lighting hindered the direct use of the captured images for 3D reconstruction, as the model’s brightness was insufficient. To address this, we used a HDR image synthesis method [18] to ensure proper brightness in the stereoscopic photos. Additionally, museum restrictions on tripods posed challenges to the photography process, prompting us to develop a VR-based stereoscopic photography guidance system. Compared to traditional photogrammetric workflows that rely on fixed shooting positions or tripod-mounted arrays, our approach enables flexible yet precise image acquisition under constrained conditions without compromising geometric accuracy.

As shown in Figure 2a, this system comprises a VR headset (Quest 3), a camera (Sony A7M4), and a gimbal. The camera is mounted on the gimbal and linked to the VR controller, which emits rays in the virtual space to guide the camera’s orientation. The stereoscopic system draws a binocular baseline in the VR display (Figure 2b,d), with the green and red double lines representing the interocular distance (60 mm, adjustable), providing a reference for the shooting direction. The photography process involves two operators (Figure 2c): one manages the camera and gimbal based on the VR guidance, while the second person takes the photo based on a sound cue from the VR system. Figure 2 outlines the procedure:Set up the green double lines.Confirm the spatial position of the green lines and display the red lines simultaneously.Roughly adjust the gimbal to bring the red guide cursor near the left scale until the white guide cursor appears.Precisely adjust the gimbal until the white cursor aligns with the green guide, and the VR system signals alignment, prompting the second photographer to press the shutter.Align the guide cursor with the right scale, and the shutter is pressed again to complete the stereoscopic photo set.Repeat this process for all four angles.

To produce HDR images, we captured five consecutive exposures at each shooting point using different shutter speeds, resulting in five sets of RAW images. For each artifact, images were captured from the left and right viewpoints at four different angles, resulting in eight synthesized HDR images (Figure 3). These high-quality inputs significantly improved texture reproduction and contributed to the overall visual realism of the final models.

During 3D reconstruction, we applied a stereo matching algorithm [28] to generate depth maps from each set of HDR stereoscopic images. After compiling depth maps from all four angles, we merged them into a complete point cloud dataset. To address potential errors and overlaps, we carefully aligned and fused the point clouds. Using a 3D reconstruction algorithm, we transformed the point cloud into a coherent 3D mesh, which was further optimized by our design team. Finally, we enhanced the visual quality by applying texture mapping, resulting in a series of high-quality 3D models of the artifacts (Figure 4).

The final models were scaled according to dimensional data provided by the museum’s publicly accessible online artifact catalog (e.g., standard measurements of diameter and height) [48]. While this approach does not guarantee absolute geometric accuracy, it ensures visual consistency with reference photographs. The scaling process was further guided by a VR-based virtual scale during image acquisition, which assisted in maintaining proportional relationships between viewpoints. While conservation policies prevented us from acquiring LiDAR ground truth data, we validated the geometric plausibility of the models through visual comparison with reference photographs and official documentation.

### 3.2. System

The replica-based virtual museum was developed using the Unity engine, with real-time multi-user interaction enabled by the Mirror library’s distributed network capability.

We conducted precise measurements of specific exhibition areas in the target museum and imported the data into Unity for 3D reconstruction, ensuring that the virtual exhibit positions and dimensions matched their real-world counterparts. We also optimized the polygon count of the 3D models to under 50k, ensuring smooth system performance.

The RVM system uses passthrough technology, allowing users to view the real world through the head-mounted display. This feature enables participants to move freely within the experimental space while wearing the headset, facilitating natural interaction with other users, thus enhancing comfort and ease of use.

For visual rendering, we employed Depth-API technology, which compares distances between real and virtual spaces to accurately depict occlusion between virtual objects and real-world elements.

We also developed a multi-user coordinate calibration module for large spaces. Unlike the ViewR system [33], we introduced a spatial anchor mechanism as a reference coordinate system for multi-user positioning. The calibration process is shown in Figure 5:The user sets the virtual ground height by touching the ground with the controller.The user selects a confirmed area on the ground as the coordinate origin using the controller’s ray.To ensure consistency, each user sets a uniform z-axis direction at the origin point. Calibration is completed once the area and direction are confirmed, establishing a spatial anchor point. This anchor ensures accurate resetting of the reference coordinates if users temporarily leave or power off their headsets.

This system enables accurate tracking and synchronization of participants’ positions even in large open spaces. Additionally, if a user’s position data are lost due to disconnection, the host system can recalibrate all participants’ coordinates to restore accurate positioning.

We use a gold mask as the virtual avatar for user synchronization. Upon entering the virtual museum, a gold mask automatically appears on each user’s face to enhance their immersive experience.

The system supports integrated deployment for both server and client, allowing standalone operation or multi-device communication over a local network. Users can designate either a personal computer as the server node or one VR headset as the host, with other devices joining as clients to enable a multi-user collaborative experience.

For user interaction, as shown in Figure 6, the system integrates Meta’s hand pose recognition module, allowing audiences to interact with and grasp virtual exhibits through natural hand poses without the need for handheld controllers. This operation enables users to better observe the details of the exhibits and allows for natural interactions related to the virtual exhibits with other users.

### 3.3. Development Cost, Time, and Scalability

The digitization process for each exhibition area involves at least one day of on-site image capture, followed by 5–10 h of 3D modeling per exhibit by a trained artist. The system was implemented in Unity using an RTX 3090 GPU, with core functionality developed within approximately one week. Its modular design supports rapid integration of new exhibits, making it adaptable for use in various heritage or exhibition settings.

## 4. Experiment

### 4.1. Experiment Design

The physical museum experiments were conducted in the Bronze exhibition areas and Sanxingdui exhibition areas. To ensure consistency in the experimental area size, we divided the bronze exhibit into two zones, labeled as Area 1 and Area 2. Participants were required to explore these designated areas freely.

RVM could be used in any open space, and to avoid external interference, the virtual tour experiment was conducted in a spacious classroom. The area was similar to the exhibition zones in the physical museum, and LED overhead lighting was installed to ensure the tracking system of the headset could operate smoothly.

The PC virtual museum, used as a reference, could be accessed on any computer. Participants engaged with the environment using keyboard (WASD) and mouse controls, navigating through the virtual space in first-person perspective. Participants only needed to run the experiment program to browse the content.

After all tests, participants completed a questionnaire. Additionally, the touring paths of participants in both the physical museum and the RVM and RVM-I museums were recorded. PC navigation paths were excluded from analysis due to fundamental operational discrepancies: Keyboard/mouse-controlled locomotion in the PC mode inherently differed from participants’ natural movement patterns in physical space, creating incomparable mobility characteristics between interface types.

The experiment received ethical approval from the university’s ethics committee.

### 4.2. Participants

We recruited 24 participants (11 male, 13 female; mean age = 25.62, SD = 6.32), predominantly university students from STEM and arts disciplines. None had previously visited the bronze exhibition hall, with 20 possessing VR experience (12 frequent VR users) and 4 being regular museum visitors. The experiment employed a cross-interval design: All participants first experienced the physical museum visit, followed by PC, RVM, and RVM-I modes with one-week intervals between each modality to mitigate interference effects. To further control for familiarity bias, participants were exposed to distinct sets of exhibits across different experimental conditions. Prior to commencement, the research objectives were explained and informed consent was obtained from all participants.

To avoid bias from viewing the same exhibits, participants were divided into three groups, with each group exploring different areas under different conditions. The detailed grouping is shown in Table 1.

The PC virtual museum, due to its different navigation method, was used only for comparison in the questionnaire, and all participants experienced it after testing other modes.

### 4.3. Apparatus and Setup

**Physical Museum Mode:** The Quest 3 headset, capable of untethered spatial positioning, was used with a simple path-recording program to track the movement path of the headsets, representing participants’ actions while observing artifacts. To avoid affecting participants’ viewing behavior, they wore the headset backward, allowing the device to be fixed without obstructing their vision, thus enabling a natural exploration of the museum. The recording program was triggered by a controller, with an experimenter responsible for controlling the recording.

**RVM and RVM-I Modes:** Four Quest 3 headsets were used as clients, with one serving as both a client and the server host. A simple UI controlled the scene content, set by the experimenter, who also managed the recording of the participants’ paths through the host device.

**PC Virtual Museum Mode:** In this mode, participants navigated using a keyboard and mouse. Multiple participants could explore simultaneously. The exhibit layout in the PC version was consistent with the real museum, and participants could move freely. When they finished their tour, the software automatically saved and uploaded their paths.

It is worth noting that the virtual environment in the PC mode was kept minimalistic—featuring only basic walls and a neutral floor—to maintain consistency with the physical environments used in the RVM condition, such as open grass fields or empty classrooms. This design choice aimed to ensure a fair comparison across different modalities by focusing participants’ attention on the exhibited artifacts rather than environmental details.

### 4.4. Procedure

As shown in Figure 7, the physical museum experiment was conducted over two days, with the time slot from 2:00 PM to 4:00 PM chosen to ensure a moderate number of museum audiences. Before starting, participants gathered at the exhibition hall entrance, where they were briefed on the designated viewing areas. Participants were divided into three groups, with each group freely touring their assigned area. Each participant spent about 10 min in the experiment, and the experiment concluded once they felt they had finished their tour. Afterward, participants were asked to fill out an initial questionnaire, which included demographic information (age, gender, museum visit frequency) and their levels of immersion and satisfaction.

The virtual museum experiment took place three days after the physical museum visit.

The RVM experiment was conducted in a spacious classroom. At the start, participants with limited VR experience were introduced to the Passthrough feature and allowed to wear the device for some time to familiarize themselves with the VR environment. Under the guidance of the experimenters, participants selected a designated area in the experimental space as the reference coordinate system origin, performing spatial calibration. The experimenters briefly explained the purpose of the experiment, namely, free exploration within the virtual museum. Each group’s experiment was conducted with three participants simultaneously, without any guidance, allowing for free interaction and movement. Once participants confirmed they were familiar with the environment, the experiment began. Each session lasted approximately 10 min.

The RVM-I experiment was conducted after the RVM session, following a 5-min break, with a process similar to RVM. At the start, participants were informed that the exhibits could be interacted with, but no instructions on operation methods were provided. Participants interacted with the virtual exhibits using natural gestures based on their own preferences.

The PC virtual museum portion followed RVM-I. Participants used a mouse and keyboard to navigate the virtual museum, which mirrored the layout of the real museum. Participants freely entered the multi-user virtual museum server through the client interface.

After the experiment, participants completed the same questionnaire as used in the physical museum, covering satisfaction, immersion, and aesthetic experience. The questionnaire also included additional sections evaluating system usability and motion sickness for the RVM and RVM-I systems.

### 4.5. Measures

#### 4.5.1. Survey

In this study, we recorded participants’ experiences in different museum environments through questionnaires, analyzing the data from six perspectives. All questionnaires used a 5-point Likert scale:

**Satisfaction (SA)**: This section combined parts of the evaluation questionnaire from [49], which measures AR systems, and questions from [50,51] on enjoyment, allowing participants to assess the content in the MU-MR digital twin museum. An example item is:”I think that learning about artifacts through physical museums\PC virtual museum\RVM\RVM-I is necessary”.

**Immersion (IM)**: This section used parts of the Player Experience of Need Satisfaction (PENS) questionnaire [52], particularly the PENS: Presence [53] and PENS: Intuitive Controls (IC) sections, combined with questions from [54] regarding immersion. Sample items are: “When moving through the virtual museum, I feel as if I am actually there”. and “When attempting to perform an action in the museum, the corresponding controls were easy to recall”.

**Aesthetic Experience (AE)**: This section used the flow-related questions from [55] to measure the overall experience of viewing the exhibits. An example item is: “I lose track of time when I view the work of artifacts”.

**Social Interaction (SI)**: This section combined social-experience-related questions from [54] and those on social quality and connection from [56]. An example item is: “I was aware of the other participant”.

**System Usability**: This section used the standard 10-item SUS questionnaire from [57] to evaluate the system usability of RVM and RVM-I.

**Motion Sickness**: Motion sickness was assessed using the standard SSQ 16-item questionnaire [58] to evaluate whether participants experienced motion sickness symptoms while using the RVM and RVM-I systems.

#### 4.5.2. Path Analysis

We analyzed the overall touring paths of participants under different conditions. In the physical museum, audiences could only observe exhibits from different angles by moving, and this behavior was directly reflected in the recorded movement data. For movement behavior, the data were broken down into three aspects: overall path distribution, time spent near the exhibits, and participants’ movement speed.

A. Path Distribution: Path data directly reflected participants’ movement during viewing. Given the large museum space and the scattered movements of viewers, we chose a 20 cm × 20 cm grid as the smallest spatial unit for analyzing the paths, presenting the data in a planar histogram.

B. Dwell Time in Proximity Area Around Exhibits: The size of the exhibits and the intricacy of their decorative patterns necessitate that audiences approach closely to appreciate these items in detail. The duration of stay is used as a key indicator to measure the effectiveness of the visitor’s tour. In the experimental exhibition areas, exhibits are similarly displayed within square glass cases with sides measuring approximately 90 cm. Given that the glass cases physically prevent audiences from getting too close to the exhibits, and taking into consideration the dimensions of the human head, we established two boundary zones based on the physical layout of museum exhibit glass cases:

(a) Nearby Area: Defined as the region within a 140 cm × 140 cm square centered around the exhibit. Audiences within this area can appreciate the detailed features of the entire exhibit.

(b) Close Area: Defined as the annular region between the 140 cm × 140 cm square and a larger 200 cm × 200 cm square centered around the exhibit. Audiences in this zone can discern the exhibit relatively clearly.

(c) Far Area: Refers to the region outside the 200 cm × 200 cm square, where it is believed that audiences cannot adequately observe the exhibit.

C. Participant Movement Speed: During the tour, participants alternated between two states:

(a) Detailed observation of exhibits, where their movement speed tended to be minimal.

(b) Moving to the next exhibit, where their movement speed increased to normal walking speed. We used participants’ movement speed in each distance range as an indicator of their viewing efficiency.

## 5. Result

### 5.1. Questionnaire Result Analysis

We used IBM SPSS Statistics 27 to analyze the questionnaire data, evaluating participants’ performance across four dimensions: satisfaction (SA), immersion (IM), aesthetic experience (AE), and social interaction (SI), based on the different modes (PM, PC, RVM, RVM-I). Since the data involved repeated measures and did not meet the normal distribution assumption, we applied the Friedman test to assess differences between the modes. The results are shown in Figure 8.

**Satisfaction** Post hoc pairwise comparisons from the Friedman test revealed significant differences in satisfaction across the modes. Satisfaction in the PC mode was significantly lower than in RVM ($\chi^{2}$ = −1.021, *p* = 0.006), PM ($\chi^{2}$ = 1.396, *p* < 0.001), and RVM-I ($\chi^{2}$ = −2.25, *p* < 0.001), indicating that the PC mode performed poorly in terms of user satisfaction. There was no significant difference in satisfaction between RVM and PM ($\chi^{2}$ = 0.375, *p* = 0.314), but the difference between RVM and RVM-I was significant ($\chi^{2}$ = −1.229, *p* = 0.001), suggesting that the interactive features of RVM-I significantly improved user satisfaction.

**Immersion** in the PC mode was significantly lower than in the physical museum (PM) ($\chi^{2}$ = 1.208, *p* = 0.001), RVM ($\chi^{2}$ = −1.208, *p* = 0.001), and RVM-I ($\chi^{2}$ = −2.25, *p* < 0.001), demonstrating that the traditional PC mode underperformed in terms of immersion. No significant difference was found between the physical museum (PM) and RVM ($\chi^{2}$ = 0, *p* = 1.000), indicating that the virtual museum provided an immersion experience comparable to that of the physical museum. However, there was a significant difference between RVM and RVM-I, with RVM-I offering a superior immersive experience.

**Aesthetic Experience** in the PC mode was significantly lower than in RVM ($\chi^{2}$ = −1.021, *p* = 0.006), PM ($\chi^{2}$ = 1.438, *p* < 0.001), and RVM-I ($\chi^{2}$ = −1.875, *p* < 0.001), indicating poor performance of the PC mode in this dimension. There was no significant difference between RVM and the physical museum (PM) ($\chi^{2}$ = 0.417, *p* = 0.264), suggesting that the aesthetic experience in the virtual museum was similar to that of the physical museum. However, the difference between RVM and RVM-I was significant ($\chi^{2}$ = −0.854, *p* = 0.022), demonstrating that RVM-I’s interactive features significantly enhanced the aesthetic experience. In contrast, there was no significant difference between PM and RVM-I ($\chi^{2}$ = −0.437, *p* = 0.240), indicating similar performance between the two in terms of aesthetic experience.

Social interaction in the PC mode was significantly lower than in RVM ($\chi^{2}$ = −0.854, *p* = 0.022), PM ($\chi^{2}$ = 1.458, *p* < 0.001), and RVM-I ($\chi^{2}$ = −1.604, *p* < 0.001), highlighting the poor performance of the PC mode in facilitating social interaction. While there was no significant difference in social interaction between RVM and PM ($\chi^{2}$ = 0.604, *p* = 0.105), the difference between RVM and RVM-I was significant ($\chi^{2}$ = −0.75, *p* = 0.044), indicating that RVM-I’s interactive features significantly enhanced social interaction compared to RVM. In contrast, there was no significant difference between PM and RVM-I ($\chi^{2}$ = −0.146, *p* = 0.696), indicating that both performed similarly in terms of social interaction.

### 5.2. SSQ and SUS

In the virtual museum (RVM/RVM-I), the participants’ average motion sickness score was 21.29, indicating that the system performed well in minimizing motion sickness, with participants reporting minimal discomfort and a generally comfortable experience.

The average usability score for the virtual museum system was 84.23. According to the SUS rating scale, this score falls within the A+ grade, indicating that participants found the system highly user-friendly, with smooth operation and an excellent overall experience.

### 5.3. User Behavior Analysis

We conducted a visual analysis of participants’ movement paths, as shown in Figure 9.

**PM:** During the visit to the physical museum, participants’ movement was restricted by glass display cases and influenced by the presence of many other audiences. Consequently, their viewing paths were widely distributed, with multiple peaks. On average, participants spent 28.18% of their time in areas distant from the exhibits and 37.84% in areas closer to the exhibits.

**RVM:** Participants were able to move more freely around the exhibits. The overall path data showed that viewers tended to cluster around the exhibits, while the pathways between exhibits are less crowded and do not see much stopping. On average, participants spent 78.64%of their time near the exhibits, while only 8.34% was spent moving along the pathways between them.

**RVM-I:** In RVM-I, participants’ movement paths were more uniform, with the majority of time spent in front of the exhibits. The overall distribution of viewing time was similar to the non-interactive mode, with participants spending an average of 79.52% of their time near the exhibits and 9.01% moving along the pathways between them.

In terms of total time, the physical museum and RVM-I exhibited similar visit durations, while RVM had a shorter overall duration. However, the time spent near the exhibits was comparable between the physical museum and RVM. In contrast, RVM-I had a significantly longer dwell time near the exhibits, with participants spending an average of 91% more time than in the physical museum and 82% more than in RVM.

### 5.4. Speed Analysis

The results are shown in Figure 10 and Figure 11.

**PM:** Participants’ movement speed remained relatively stable across all distances. The average speed near exhibits was 0.268 m/s (SD = 0.048, median = 0.228), while the average speed between exhibits was 0.414 m/s (SD = 0.088, median = 0.400).

**RVM:** In the “Nearby” area, participants moved slightly faster than in the physical museum, with an average speed of 0.381 m/s (SD = 0.052, median = 0.373). The speed between exhibits increased significantly, averaging 0.682 m/s (SD = 0.053, median = 0.690).

**RVM-I:** In this mode, participants’ speed near exhibits dropped significantly, averaging 0.186 m/s (SD = 0.053, median = 0.065). The speed between exhibits was 0.516 m/s (SD = 0.113, median = 0.587).

The results of pairwise comparisons following Friedman tests on the speed distribution across different modes are presented. Apart from the insignificant difference observed between the physical museum (PM) and RVM modes in far-distance areas (*p* > 0.05), all other comparisons showed significant differences.

In the Nearby region, there was a significant difference in visitor movement between PM and RVM ($\chi^{2}$ = 0.384, *p* < 0.001), as well as PM and RVM-I ($\chi^{2}$ = 0.822, *p* < 0.001). A significant difference was also found between RVM and RVM-I ($\chi^{2}$ = 0.438, *p* < 0.001).

In the Far region, there was a significant difference in visitor movement between PM and RVM ($\chi^{2}$ = 0.522, *p* < 0.001), as well as PM and RVM-I ($\chi^{2}$ = −0.066, *p* = 0.091). Additionally, a significant difference was noted between RVM and RVM-I ($\chi^{2}$ = 0.457, *p* < 0.001).

For the entire area, significant differences were observed between PM and RVM ($\chi^{2}$ = 0.389, *p* < 0.001), as well as PM and RVM-I ($\chi^{2}$ = 0.292, *p* < 0.001). There was also a significant difference between RVM and RVM-I ($\chi^{2}$ = 0.681, *p* < 0.001).

These findings indicate that both RVM and RVM-I modes have a substantial impact on participants’ browsing patterns, altering certain aspects of their visiting behavior.

## 6. Discussion

### 6.1. Objective Evaluation

Significant differences were observed in effective experience time across the different modes. The proportion of effective experience time ranked as RVM-I ≈ RVM > PM. In the RVM-I and RVM modes, effective experience time nearly doubled compared to the PM group, while movement efficiency remained consistent across all modes. This suggests that crowding in the physical museum had a considerable impact on viewing efficiency, as participants’ direct paths to exhibits were often blocked, and popular artifacts were surrounded by participants, leading to wait times before approaching displays. In terms of overall effective experience time, the ranking was RVM-I > RVM ≈ PM, indicating that RVM and PM were similarly engaging for participants, though RVM required significantly less movement time than PM. The increase in effective experience time in RVM-I is attributed to the interactivity of the exhibits, allowing viewers to appreciate the exhibits from multiple angles in a more natural way and discover more easily overlooked details, which directly led to a significant increase in effective experience time.

Visualized movement paths and speed distribution diagrams showed that participants in the RVM mode exhibited uniform movement speeds, consistent with behavior in real museums. In the RVM mode, participants circled the virtual artifacts during viewing, while in the PM mode, taller peaks in movement data were observed, largely due to crowding, representing waiting times and locations. The interactive features of RVM-I transformed the way participants engaged with artifacts. Without physical barriers like glass display cases, most participants reached out to touch the virtual artifacts. When able to freely handle the items, participants abandoned the traditional walking-around approach, instead opting to rotate and examine the objects by hand while maintaining a stationary posture. The increased peaks in RVM-I indicate that participants were closely examining artifact details.

### 6.2. Subjective Evaluation

The traditional PC-based virtual museum received the lowest overall rating when using the same exhibits, showing significant differences compared to all other modes.

Among PM, RVM, and RVM-I, satisfaction was highest for RVM-I, with a notable difference compared to RVM. Although the exhibits in PM featured the most refined shapes and textures, the crowd reduced audience satisfaction in terms of cultural engagement. Participants were distracted by the crowd and its diverse behaviors, which interfered with their ability to fully appreciate the artifacts and absorb cultural information. While RVM’s exhibit detail was slightly lower than PM’s, it avoided the disruptive effects of large crowds. Despite the absence of interactivity, participants showed more focused engagement with the artifacts, as supported by behavioral data presented in Section 5.3. In RVM-I, the interactive features enhanced exhibit appeal and increased perceived authenticity, helping to offset the impact of reduced visual detail compared to PM.

Immersion is an experience that varies based on the environment—real spaces evoke immersion in physical environments, while virtual spaces elicit immersion in digital settings. The browsing experiences in RVM and PM were similar. Although RVM provided participants with a more convenient experience by shielding them from large crowds, the atmosphere of PM, which naturally isolates external distractions, still offered a high level of immersion. Consequently, the immersion levels between the two were largely comparable. In RVM-I, participants were fully engaged with the exhibits through interaction, enhancing their sense of authenticity toward the virtual elements and further increasing immersion.

Regarding aesthetic experience, there were no significant differences between PM and RVM, suggesting that virtual museums have essentially matched real museums in terms of artifact shapes, textures, movement patterns, and multi-user viewing experiences. However, there were significant differences between RVM and RVM-I. In RVM-I, participants could pick up artifacts and view them from any angle, which was more time-efficient and physically easier than moving around the exhibits, providing greater freedom. The interactive breakdowns of the artifacts were also engaging. Participants remained stationary during viewing, significantly reducing cognitive load and thereby improving aesthetic efficiency in RVM-I.

In terms of social experience, PM and RVM-I were similar and rated higher than RVM. As a real-world activity, PM offers an unmatched advantage in terms of sociability. In RVM, participants wore headsets, which somewhat hindered the social experience by preventing non-verbal communication such as facial expressions. Additionally, only users within RVM could share information about the virtual exhibits, while others in the same physical space were unable to see the virtual world, further limiting the social interaction in RVM. The artifact interaction in RVM-I facilitated greater communication between participants. Users expressed considerable interest and enthusiasm in the artifact-carrying and passing features of RVM-I, frequently picking up artifacts and sharing knowledge and insights with companions. This significantly enhanced social interaction compared to RVM.

We also explored other aspects of participants’ experiences. Most participants expressed a preference for arranging artifacts in the virtual museum. In RVM-I, both male and female participants engaged in spontaneous and diverse interactive behaviors, such as gathering artifacts together for comparative analysis. Notably, participants expressed interest in additional interactive features (which are part of another work, as shown in Figure 12a,b), such as interactive tasks and gamified functions. Examples of such features included artifact treasure hunts, fragment stitching, the interactive “Houyi Shoots the Suns” myth, and a bronze artifact pitching game.

In conclusion, we can conclude that RVM can partially substitute physical museums, while the RVM-I alters the visitor experience by introducing an interactive model that enhances the audience’s overall engagement.

### 6.3. Limitation and Future Work

This study and system also have several limitations:The current virtual museum system can operate indoors and in outdoor areas under overcast weather conditions. However, exposure to direct sunlight can damage the VR sensors and display screens, so the system is not recommended for use outdoors in sunny conditions.The spatial anchor system of the system is still not sufficiently refined. The calibration module requires each user to individually calibrate the reference coordinate system. Additionally, in specific, almost entirely textureless spaces—which are very rare—spatial coordinates may experience slight drift, leading to misalignment of user coordinates in the virtual world.The depth sensor’s precision is not perfect, causing excessive occlusion for distant areas due to detection accuracy. Additionally, there may be occlusion errors when used outdoors during the day or in environments with low light levels, which can also affect precision.The participant pool was imbalanced in terms of both prior VR experience and museum visiting habits, with only 4 out of 24 participants considered new to VR and a minority identifying as frequent museum visitors. Although this limited our ability to perform subgroup comparisons, we found that overall trends in user feedback and behavioral data (e.g., dwell time, questionnaire responses) were consistent across participants. Nevertheless, future studies will aim to address this limitation by implementing more controlled participant recruitment strategies—particularly balancing groups based on VR familiarity and museum visit frequency. Building on this, we will also incorporate additional human–computer interaction (HCI) elements—such as gesture-based navigation, voice interaction, and personalized guidance systems—to examine how different user groups interact with and experience digital cultural heritage systems. This approach will help us better understand the usability, accessibility, and engagement differences across diverse audiences. We will also include qualitative data, such as interviews and open-ended responses, to better understand users’ personal experiences.

In our future plans, accessibility services are becoming increasingly important in museums, as individuals with visual, auditory, physical, or intellectual disabilities also have the right to appreciate art. The RVM-I proposed in this study holds potential for use in accessible museums. People with physical disabilities can experience the shapes and patterns of famous artifacts from domestic and international museums from their local communities, addressing challenges such as travel difficulties and high costs associated with museum visits. We plan to continue upgrading the RVM-I to address issues such as viewing barriers created by glass cases, aiming to enhance guidance features and exhibit layouts to meet the needs of individuals with physical and visual impairments. (as shown in Figure 12c–f).

In addition, we will explore how different spatial contexts—such as historically accurate reconstructions, fantasy-inspired environments, or thematic exhibitions—may influence user engagement, presence, and learning outcomes. This will help us understand whether specially designed environments can lead to improved immersion and overall satisfaction.

To gain deeper insights into these effects, we will combine quantitative measures (e.g., tracking data, physiological responses) with qualitative methods—such as post-experience interviews and focus group discussions—to evaluate subjective experiences that cannot always be captured through numerical data alone.

Furthermore, building on the findings of this study, we aim to conduct more controlled experiments in future work by categorizing participants based on their prior VR experience and museum visiting habits. By comparing subgroups such as VR-experienced vs. non-VR users, or frequent vs. infrequent museum visitors, we hope to better understand how background factors affect interaction patterns, usability, and emotional connection with digital cultural heritage.

In particular, qualitative data will play a key role in uncovering subtle differences in perception, preference, and emotional resonance among these groups.

## 7. Conclusions

In our study, key findings have surfaced from our results. First, RVM and RVM-I can be deployed in any indoor or outdoor open space, avoiding large crowd gatherings and significantly enhancing the efficiency of cultural artifact exhibitions. Both modes are similar to PM in terms of aesthetic experience and social interaction. RVM and RVM-I maintain consistency with PM in visual presentation and exhibition format, providing the audience with a temporal and spatial experience akin to that of the real world.

Second, while audiences typically appreciate music, fine art, or films from a seated or fixed standing position, the three-dimensional structure of bronze artifacts requires a circumferential viewing approach, which can diminish the audience’s appreciation experience. RVM-I offers a stationary viewing option, changing the traditional circumferential observation method and reducing physical exertion, which aligns with the principle of minimal effort in aesthetic experience. Compared to RVM, RVM-I enhances the quality, efficiency, and effectiveness of aesthetic appreciation, providing a more immersive experience.

Additionally, RVM-I provides interactive features, further improving user satisfaction and immersion. The large-space multi-user museum, which employs spatial anchor technology, enables the rapid dissemination of digital artifacts, allowing audiences to enjoy traditional art more conveniently and freely, while also encouraging participation in artistic interactions and the exploration of cultural treasures. 

## Figures and Tables

**Figure 1 sensors-25-04046-f001:**
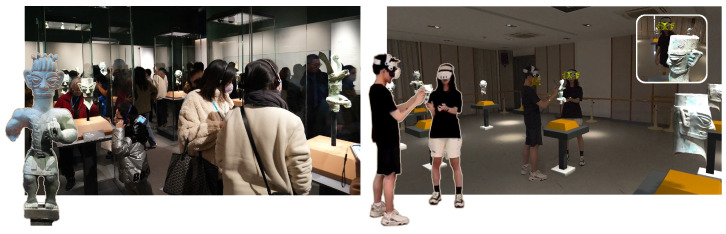
(**Left**) Physical Museum (**Right**) Replica-based Virtual Museum.

**Figure 2 sensors-25-04046-f002:**
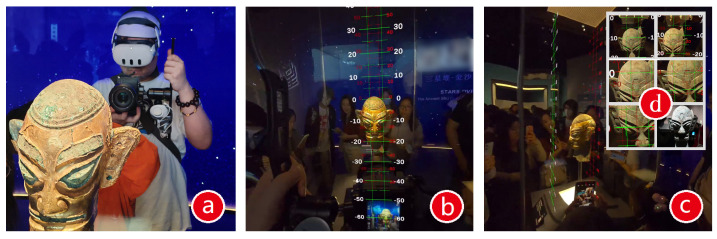
The VR-based stereoscopic photography guidance system. Image (**a**) shows an image taken during device operation; images (**b**–**d**) depicting real screens from the system.

**Figure 3 sensors-25-04046-f003:**
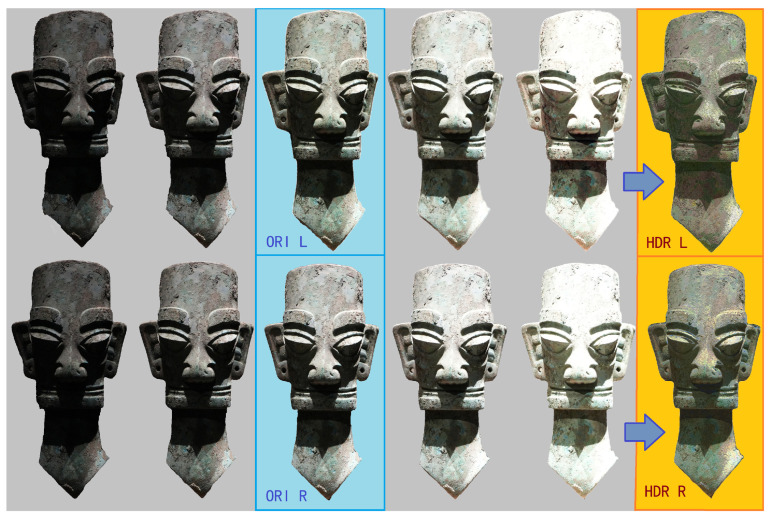
HDR stereoscopic photos were obtained by combining multi-level exposed images at fixed positions.

**Figure 4 sensors-25-04046-f004:**
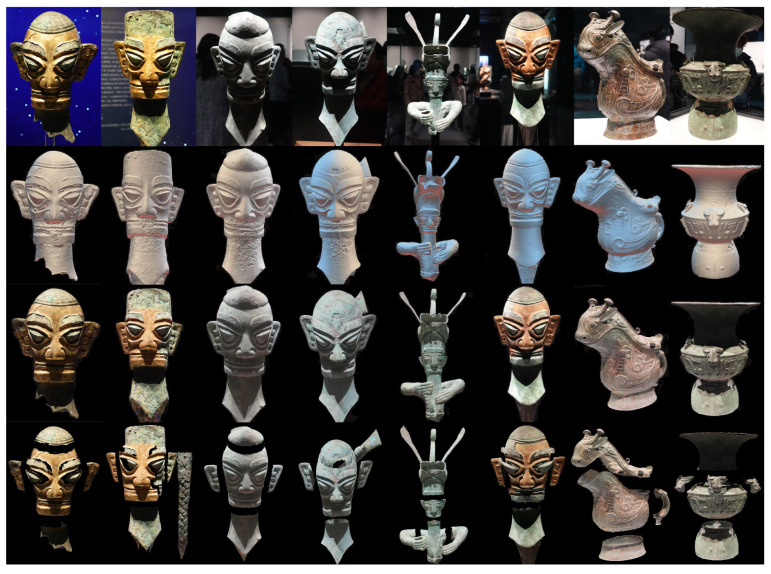
3D artifact model creation (Top row: photos of real artifacts; second row: 3D reconstruction base models; third row: 3D textured models; fourth row: models broken down for subsequent interactive content creation).

**Figure 5 sensors-25-04046-f005:**
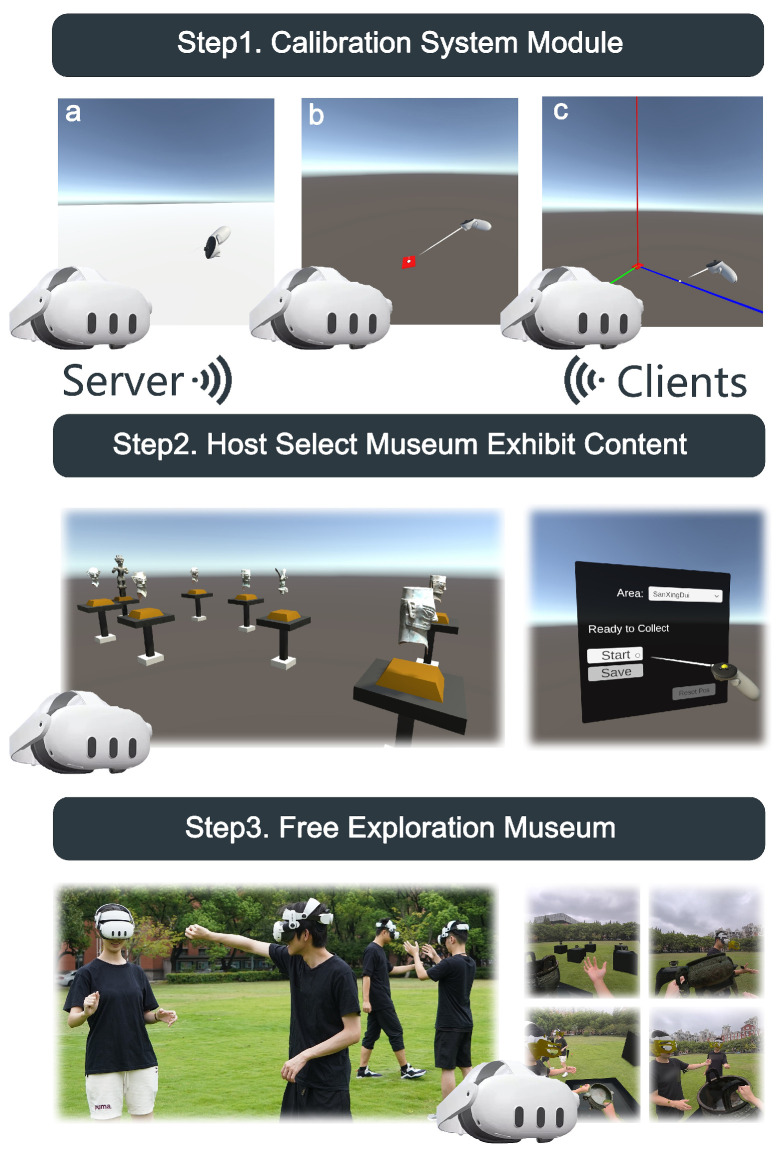
System usage scenarios. All participants first perform spatial calibration, then select whether to be the host or a client, and communicate via Wi-Fi. The host can configure the exhibition content, enable path recording, and reset participants coordinates. Once set, all HMDs can freely navigate the scene.

**Figure 6 sensors-25-04046-f006:**
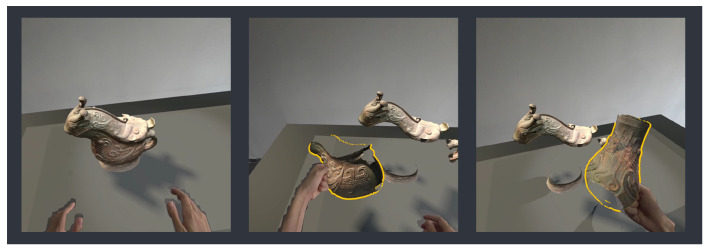
As shown in the figure, users can use natural grasping gestures to pick up different parts of the exhibit for closer inspection.

**Figure 7 sensors-25-04046-f007:**
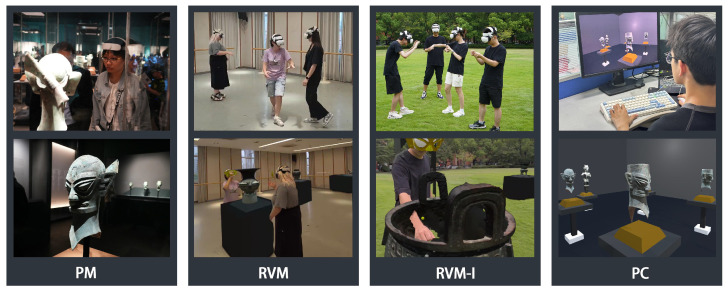
This figure illustrates the system’s state under various modes. For each mode, the first image represents the participant’s status within their environment, while the second image reflects the content visible from the participant’s perspective.

**Figure 8 sensors-25-04046-f008:**
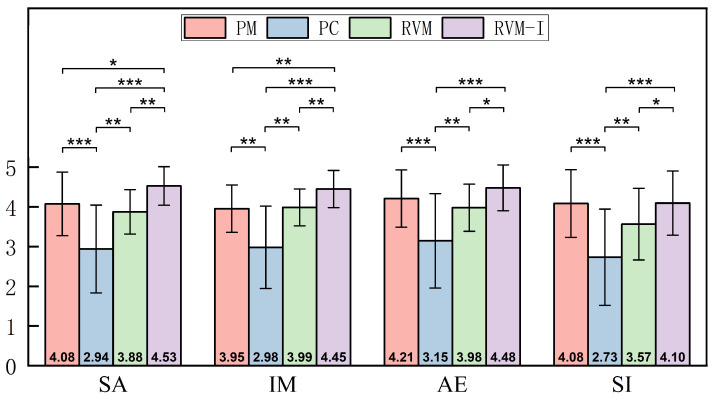
The mean and standard deviation of each dimension under different modes. The significant differences between paired comparisons are marked in the figure (**p* < 0.05, ** *p* < 0.01, *** *p* < 0.001).

**Figure 9 sensors-25-04046-f009:**
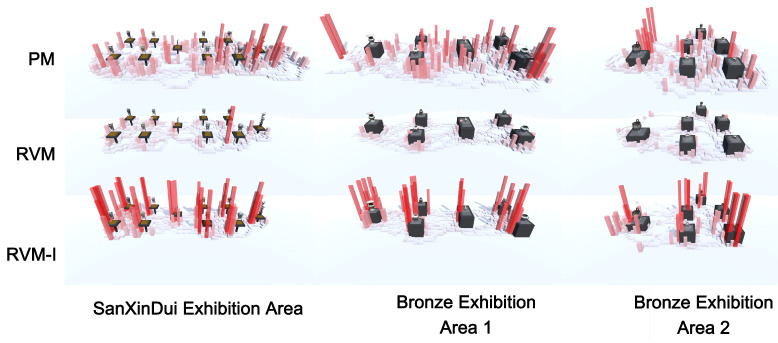
Visualization of participants’ browsing paths, from top to bottom: physical museum (PM), RVM, and RVM-I. From left to right: Sanxingdui exhibit area, Bronze Exhibit Area 1, and Bronze Exhibit Area 2. In the PM mode, the presence of glass display cases necessitates that the audience maintain a distance from the exhibits. In RVM and RVM-I modes, the removal of the glass barrier allows the audience to appreciate the exhibits more closely. This change also results in parts of the path visualization content being obstructed by the display case models.

**Figure 10 sensors-25-04046-f010:**
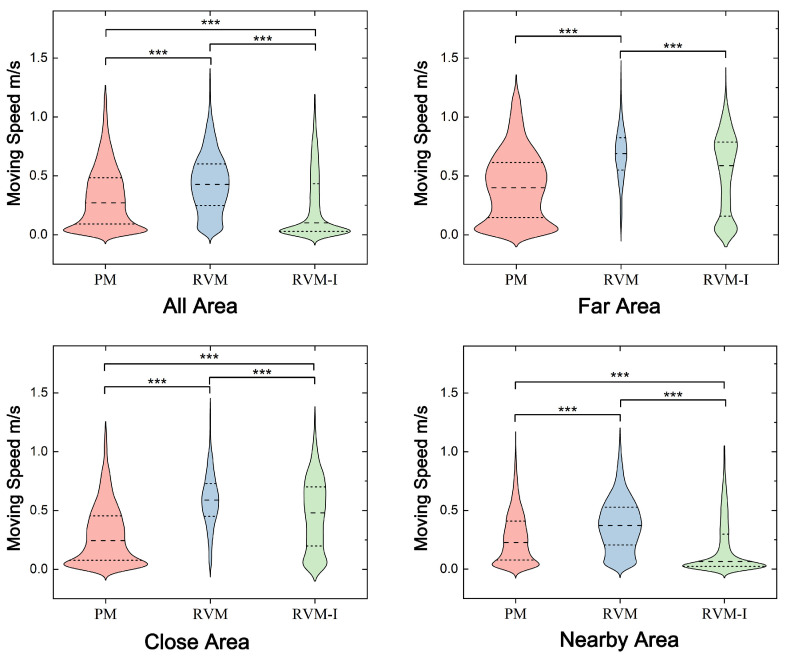
Distribution of average speed and dwell time across different modes. The significant differences between paired comparisons are marked in the figure (*** *p* < 0.001).

**Figure 11 sensors-25-04046-f011:**
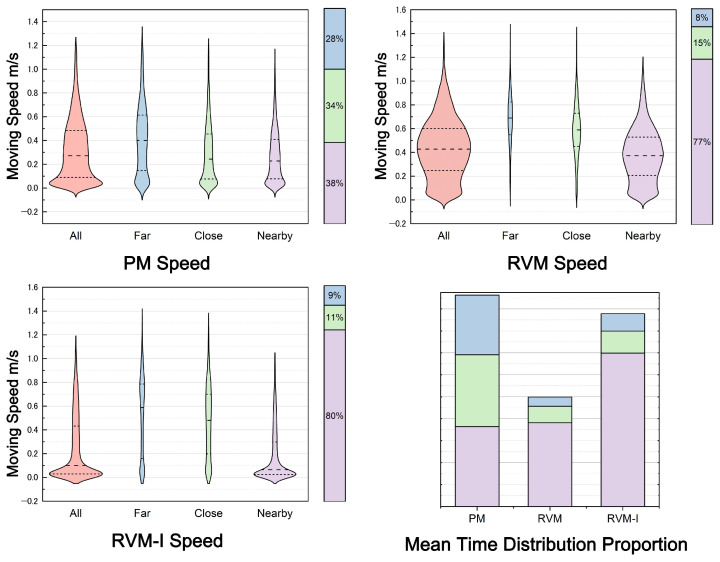
Speed comparison across modes and distances.

**Figure 12 sensors-25-04046-f012:**
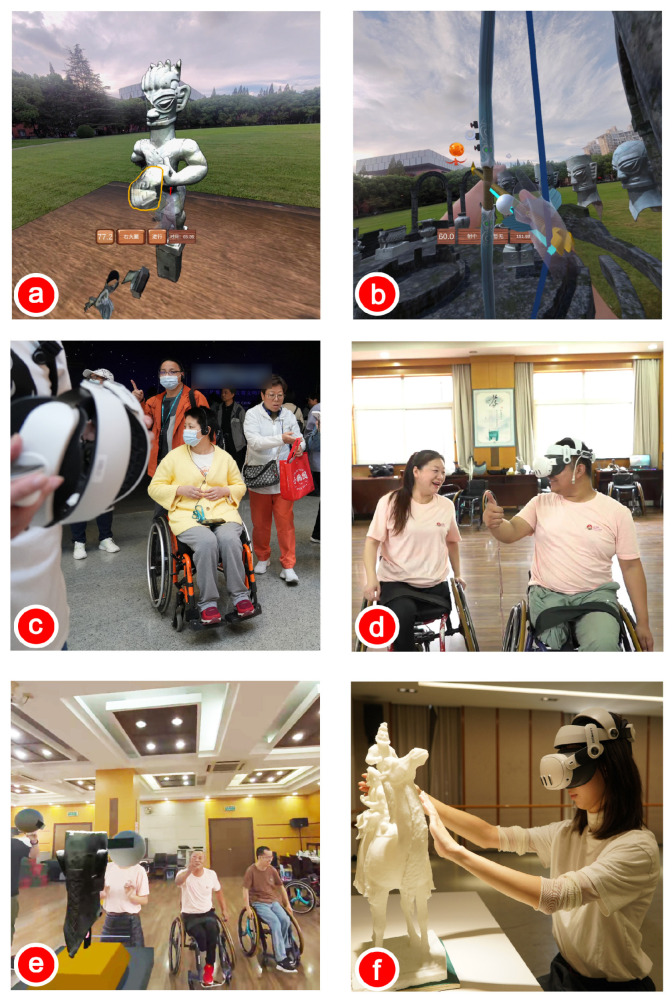
The future expansion direction of RVM will target more special groups of people and enhance interactive fun, helping more people to accept the information behind cultural relics. (**a**,**b**) Games; (**c**,**d**) accessibility for physically disabled; (**e**) multi-user system; (**f**) touch-based museum interface.

**Table 1 sensors-25-04046-t001:** Division of areas for each group.

	First Group	Second Group	Third Group
PM	Sanxingdui exhibition areas	Bronze exhibition Area 1	Bronze exhibition Area 2
RVM	Bronze exhibition Area 2	Sanxingdui exhibition areas	Bronze exhibition Area 1
RVM-I	Bronze exhibition Area 1	Bronze exhibition Area 2	Sanxingdui exhibition areas

## Data Availability

Data are contained within the article.

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
