# Peer review of "Contrasting Physical and Virtual Museum Experiences: A Study of Audience Behavior in Replica-Based Environments"

_sensors, 2025, doi:10.3390/s25134046_

Round 1
Reviewer 1 Report
Comments and Suggestions for Authors
My suggestion to the authors is to organize participants, in further experiments, into equal groups according to familiarity with VR. I think it will show a more precise comparison between the group that did not have prior experience with VR and the group that had previous experience. This can also be applied to the experience of visiting the museums. It would be interesting to see if the experience will vary between the participants who are regular museum visitors and the ones who aren't.
Section 4.3. Apparatus and Setup: According to the provided images, PC Virtual Museum Mode is designed relatively simply and does not resemble the real-world environment, hence my opinion why the engagement (overall results) with this setup was poor in comparison to others. It would be interesting to see if a variation to the environment (realistic, fantasy-like, put in some context related to the historic period of the exhibited artifacts, etc.) in the PC Virtual Museum Mode would lead to improved results.
Page 5, Line 156: Correct the small case letter at the beginning of the Paragraph, to capital case.
Reviewer 2 Report
Comments and Suggestions for Authors
The paper is interesting, uses good methods and results are clearly presented. Yet there are some points which can be further elaborated to improve the quality of the paper. Please take the comments below into account. The purpose is not to discourage the authors but rather to improve the quality of the paper and overall contribution of the research.

Round 2
Reviewer 2 Report
Comments and Suggestions for Authors
Thank you for working to improve the paper's readability, conciseness and contribution. Your edits provide a better idea about the purpose, development, evaluation and contribution of RVM and particularly RVM-I systems for audience engagement with heritage settings.
While acknowledging recent review/edits to your work, I would propose the following minor corrections, before publication:
- You mention that “multi-user mixed reality (MU-MR)” appears in full with acronym following upon its first appearance in Section 1 (line 70). Maybe this was forgotten, as the full words do not appear in line 70 (or elsewhere).
- In lines 39-41 you refer to real world problems to better contextualise and provide a rationale for the research, yet it would be good to support such claims (practical challenges in visiting museums) by adding some references.
- In 6.3 where you explain future research plans, to explicitly mention the inclusion of qualitative data to evaluate aspects of the experience where quantitative data might not be enough. This is implied in paragraph 4 but might not be easily understood.
